# Show Me a Safer Way: Detecting Anomalous Driving Behavior Using Online Traffic Footage

**Xiao Zheng, Fumi Wu, Weizhang Chen, Elham Naghizade * and Kourosh Khoshelham** 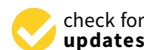

Department of Infrastructure Engineering, University of Melbourne, Parkville, VIC 3010, Australia;
xzheng5@student.unimelb.edu.au (X.Z.); fumiw@student.unimelb.edu.au (F.W.);
weizhangc@student.unimelb.edu.au (W.C.); k.khoshelham@unimelb.edu.au (K.K.)
* Correspondence: enaghi@unimelb.edu.au

**Abstract:** Real-time traffic monitoring is essential in many novel applications, from traffic management to smart navigation systems. The large number of traffic cameras being integrated into urban infrastructures has enabled efficient traffic monitoring as an intervention in reducing traffic accidents and related casualties. In this paper, we focus on the problem of the automatic detection of anomalous driving behaviors, e.g., speeding or stopping on a bike lane, by using the traffic-camera feed that is available online. This can play an important role in personalized route-planning applications where, for instance, a user wants find the safest paths to get to a destination. We present an integrated system that accurately detects, tracks, and classifies vehicles using online traffic-camera feed.

**Keywords:** object detection; vehicle tracking; driving-behavior classification

## 1. Introduction

Traffic management aims at reducing traffic incidents and improving traffic flow. Malicious or unintentional anomalous driving behaviors, as categorised and shown in Figure 1, can directly or indirectly result in traffic incidents and affect transport efficiency [1]. Consequently, knowing where, when, and how often these behaviors happen is highly valuable both to road users, e.g., to adjust their path through safer routes, and road authorities, e.g., to improve signage. Traditional hardware-based detection techniques, such as inductive loop detectors, laser detectors, and optical detectors, are expensive to install and maintain [2]. Moreover, these techniques have limited functionalities when it comes to analyzing the behavior of drivers and other road users. For instance, an inductive loop detector cannot distinguish a cyclist from a car.

Traffic cameras have been widely used to monitor traffic conditions, contributing to urban surveillance [3] and intelligent transportation systems [4]. Compared with traditional sensors, widely installed traffic cameras are more cost-effective and have a wider field of view, thus allowing to monitor multiple lanes [5] and to distinguish between road users. With advances in computing methodologies, in particular in the field of computer vision, as well as the development of more affordable and high-performance cameras, vision-based vehicle detection and tracking is now becoming more feasible, hence offering great potential [6] for automated traffic-information collection and analysis [7].

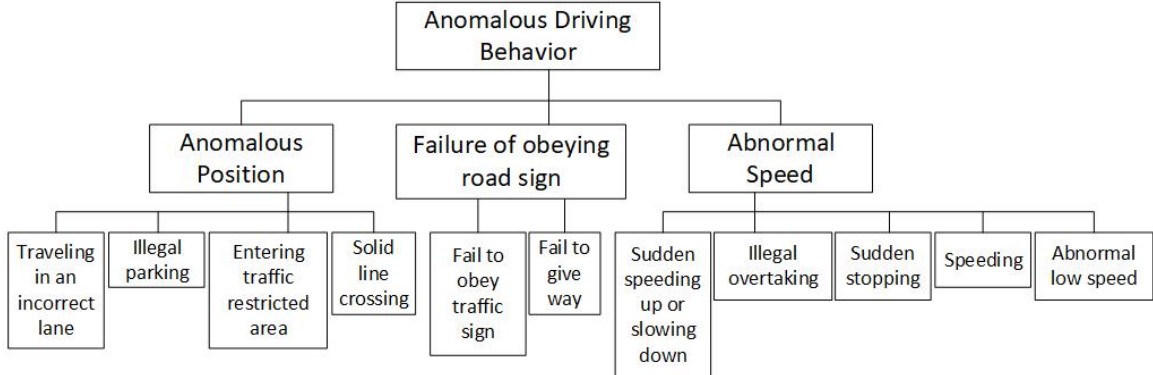

**Figure 1.** Classification of typical anomalous driving behaviors.

While research in this area has mainly focused on vehicle detection [8], vehicle tracking [9] and understanding road users' behavior [10] separately and on an ad hoc basis, there is a lack of research on integrated systems that combine detection, tracking, and driving-behavior classification using traffic footage collected from online traffic cameras. To address this issue, we propose a web-based integrated system for the detection and analysis of anomalous driving behaviors, which can potentially be used to provide tailored safe-route recommendations for drivers and other road users. Our system can also provide information to traffic-management authorities in order to help them make better decisions.

Our system uses traffic footage collected from online sources such as Live Traffic Cameras in Australia (https://straya.io/), and state-of-the-art visual-recognition method Mask R-CNN, which is fast, simple, and provides accurate results [11], to first detect the vehicles on the road. The system then tracks the detected vehicles by using a combination of the Hungarian algorithm and Kalman filter [9]. The trajectories of the tracked vehicles are analyzed to determine anomalous driving behaviors, including speeding, crossing a solid line, and entering traffic-restricted areas. The spatiotemporal spread of anomalous driving behaviors is then communicated to road users in the form of hotspot maps to enable them to tailor their routes.

## 2. Related Work

### 2.1. Vehicle Detection

Vehicle detection aims at identifying vehicles and their locations in images. Two main domains of vehicle detection are motion-based approaches [12], detecting vehicles based on their difference with static backgrounds, and appearance-based approaches [8], using prior knowledge to segment the foreground and background using manually designed features such as color [13], texture [14], and Histogram of Gradient (HOG) [15]. By contrast, features extracted by Convolutional Neural Networks (CNNs) and other methods based on CNNs are automatically learned [16]. A Faster Region-based CNN (Faster R-CNN) extracts features on a number of proposal regions, and then performs classification and bounding-box regression to detect various objects including vehicles [17]. Combined with a Region Proposal Network (RPN), which proposes candidate bounding boxes, Faster R-CNN can efficiently perform both hypothesis generation and verification [18]. As shown in Figure 2, compared with Faster R-CNN, the Mask R-CNN preserves pixel-level locations to improve mask accuracy, and then adds a branch for predicting segmentation masks on each Region of Interest (ROI), in parallel with a branch for classification and bounding-box regression. The Mask R-CNN achieves state-of-the-art performance in instance segmentation and object detection, with a fast training and testing speed [11].

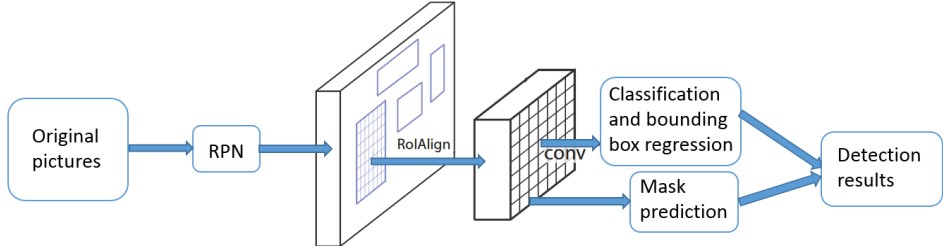

**Figure 2.** Mask Region-Convolutional Neural Network (R-CNN) framework.

*2.2. Vehicle Tracking*

Vehicle tracking aims to re-identify vehicles, derive vehicle trajectories, and predict vehicle positions and motions based on the patterns of previous positions and motions [9]. The methods of video-based tracking can be systematically classified as a contour-based method establishing the outline of moving objects and updating outlines according to the motion of an object [19]; feature-based methods concentrating on tracking significant features of the vehicle and the distribution of these features [10]; and framework-based methods in which objects in a framework are assumed to move in accordance with certain distribution patterns like position and velocity [20].

The Hungarian algorithm [21] is an optimization algorithm used to solve the assignment problem that deals with the allocation of resources to activities, seeking for minimum cost or maximum profit. The authors in Reference [20] used a Kalman filter to predict the centroid of the observed vehicle on successive frames, calculate measurement errors and prediction errors, and dynamically update predictions. Our system shows that an ensemble method that integrates these two approaches provides higher accuracy.

*2.3. Behavior Classification*

Behavior classification aims at characterizing the behavior of drivers as normal or abnormal, and identifying anomalous events such as sudden stopping and solid-line crossing. The authors in Reference [10] classified sudden stopping behavior by setting thresholds for vehicle starting velocity, ending velocity, and accelerated speed, and if the corresponding three values of the observed vehicle in the real world are smaller than the set thresholds, the behavior of the observed vehicle can be judged as sudden stopping. For the classification of solid-line crossing, Reference [10] specifies solid lines as boundary-referenced lines, extracts their pixel coordinates, and then computes the variance of the differences between the coordinates of vehicle trajectories and the solid referenced line. If the vehicle keeps moving within a lane, the variance stays at a correspondingly low level, whereas if the vehicle crosses the solid line to change the lane, there is a sudden dramatic increase in the variance value.

However, these studies only focused on one or a few specific vehicle behaviors, which limits their application in complex traffic situations. In summary, there is a lack of an integrated system that combines detection, tracking, and classification to identify anomalous driving behaviors based on traffic footage.

**3. Methods**

To create an integrated system that can be used to detect anomalous driving behaviors based on traffic footage, we performed vehicle detection, tracking, and anomalous-driving-behavior classification using the steps as explained below.

### 3.1. Vehicle Detection

To detect the vehicles in the frames, we used the Mask R-CNN technique. As shown in Figure 2, Mask R-CNN uses an RPN to generate candidate-object bounding boxes and RoIAlign to extract features and preserve pixel-level locations to improve the accuracy of the mask. The network includes a branch for predicting segmentation masks on each ROI, in parallel with a branch for classification and bounding-box regression. This method is both fast and simple, and provides accurate results in instance segmentation and bounding-box object detection [11]. Video footage is converted to a sequence of images, and fed as input to the Mask R-CNN. With pretrained weights for MS COCO [22] which can detect a wide range of objects (for instance, cars, buses, and pedestrians), the resulting class ID, frame number, confidence level, and the coordinates of the bounding boxes for each detected vehicle are collected and recorded. Vehicle centroids can be calculated as centroids of corresponding bounding boxes. Figure 3 shows an example of vehicle detection in an image frame using Mask R-CNN.

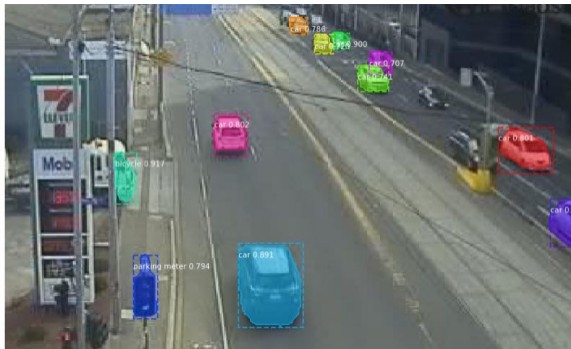

**Figure 3.** Vehicle detection using Mask R-CNN.

### 3.2. Vehicle Tracking

Once a vehicle is detected, vehicle tracking aims to determine centroids belonging to the same vehicle in continuous frames and accordingly derive the trajectories of the vehicles. The Hungarian algorithm assigns the centers of vehicles to the trajectories of those vehicles with the least total cost, which is the sum of distance between centers and trajectories, with the condition that one center is exclusively assigned to one trajectory.

To eliminate the influence of vehicle-detection errors on the Hungarian algorithm, a Kalman filter is used that can predict the vehicle centroid location in case it is missed by the detection method. The complete tracking process is shown in Figure 4 and comprises the following steps.

1.  Newly appearing vehicles are identified, and tracks are created with detected centroids. When the distances between detected centroids and the last centroids in all tracks exceed a predefined threshold, this detected centroid is considered as a newly appearing vehicle.
2.  The Hungarian algorithm is used to assign the rest of detected centroids to existing tracks by minimizing the sum of cost between assigned detected centroids and predicted centroids in tracks.
3.  A Kalman filter is used to predict and correct the detected centroids by estimating the centroid of next frame based on the given centroid of the current frame.
4.  Remove and extract existing trajectories when there is no corresponding vehicle being assigned to these tracks for five frames (one second), which indicates that the vehicle has moved out of the camera range.

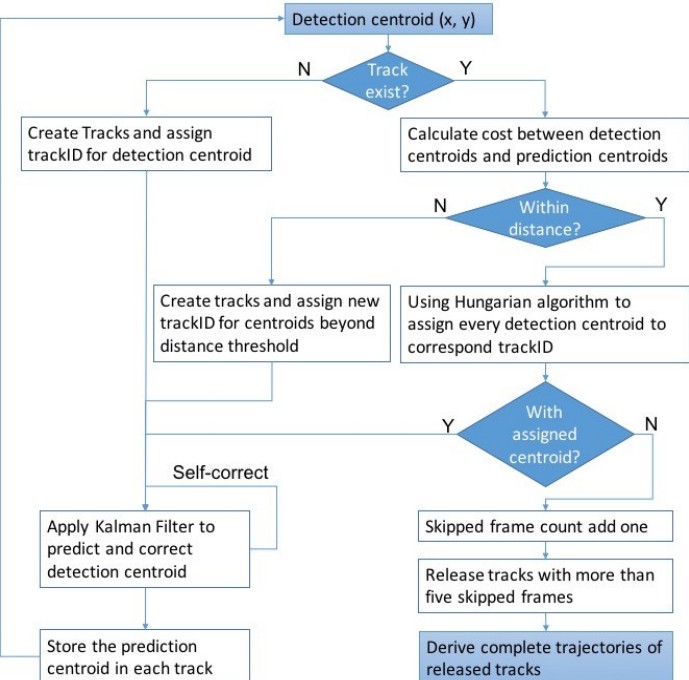

**Figure 4.** Tracking process.

### 3.3. Behavior Classification

Behavior classification aims to determine and classify anomalous driving behaviors, and is mainly based on trajectory information obtained in the vehicle-tracking component. Here, we classify three common anomalous driving behaviors as follows:

1.  Speeding: Identified by a comparison between the speed limit and the detected average velocity of the observed vehicle while traveling in a specified region. As shown in Figure 5, two green boundary lines are set for recording where and when a vehicle enters and leaves the region. The positions and frame numbers of a vehicle are recorded once the vehicle passes the upper green line or the lower green line, and time is correspondingly recorded. To correct the perspective effect of a traffic camera, a projective transformation [23] is used to map every point in the footage to its real-world coordinates on the ground plane. For speed estimation, Equation (1) is used:

$$v = \frac{s}{t} = \frac{\sqrt{(x_j - x_i)^2 - (y_j - y_i)^2}}{t_j - t_i} \tag{1}$$

where $v$ is the speed, $s$ is the distance that the vehicle traveled between the upper green line and the lower green line area, $(x_i, y_i)$ and $(x_j, y_j)$ are the centroid coordinates of vehicles on the ground plane, and $t_i$ and $t_j$ are the traveling time between entering the $i$th frame and leaving the $j$th frame.

To detect anomalous speed above or below the speed limit, we define an uncertainty margin for the estimated speed. The uncertainty of the estimated speed is calculated based on the theory of variance propagation, using the uncertainty of traveled distance $s$ and travel time $t$. Assuming that distance and time are mutually independent, we have:

$$\sigma_v^2 = (\frac{1}{t}\sigma_s)^2 + (\frac{s}{t^2}\sigma_t)^2 \tag{2}$$

where $\sigma_s$ and $\sigma_t$ are the standard deviation of traveled distance and travel time, respectively, which are obtained from the position of the detected vehicle in multiple consecutive frames. Specifically,

the standard deviation of distance is computed by measuring the distance traveled between two consecutive frames across *n* frames. For the standard deviation of travel time, a constant value corresponding to $\frac{1}{2}$ a time interval between consecutive frames is assumed.

2.  Solid-line crossing: Recognized by establishing whether the trajectory of a detected vehicle intersects with a buffer created around the solid line (two white lanes in Figure 5). If the centroid of the detected vehicle is within the created buffer, it is determined as crossing the solid line.

3.  Entering traffic-restricted areas: This behavior is identified by the overlapping area between the traffic-restricted area shown as the hatched red polygon in Figure 5, and a circular buffer with a predefined radius created around the vehicle centroid. If the overlapping area is larger than a specific percentage of the circular buffer area, i.e., approximately at least half of the car is within the traffic-restricted area, the vehicle is determined to have entered the traffic-restricted area.

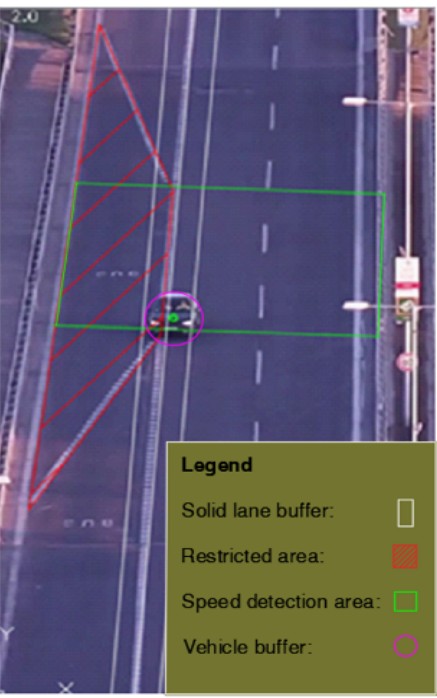

**Figure 5.** Manually specified regions used for behavior classification.

## 4. Experiments

### 4.1. Experiment Setup

We used the following thresholds for behavior classification:

- For speeding, we considered an uncertainty margin $v \pm \sigma_v$ that corresponds to a confidence interval of 68%.
- For entering traffic-restricted areas, if the overlap area was larger than 50% of the circular buffer area, the vehicle was determined to have entered the traffic-restricted area.

To evaluate the performance of our integrated system for detecting solid-line crossing and entering traffic-restricted areas, recall, precision, and false positives per second were utilized as a measure of detection accuracy. These three coefficients are defined as:

$$Recall = \frac{N_{\text{TP}}}{N_{\text{TP}} + N_{\text{FN}}} \tag{3}$$

$$Precision = \frac{N_{\text{TP}}}{N_{\text{TP}} + N_{\text{FP}}} \tag{4}$$

where $N_{TP}$ is the number of True-Positives (TP), which is the number of anomalous behaviors that are correctly detected; $N_{FN}$ is the number of False-Negatives (FN), which is the number anomalous behaviors that could not be detected by the system; and $N_{FP}$ is the number of False-Positives (FP), which is the number of anomalous behaviors that do not actually happen but are wrongly detected.

In our system, we used two sets of traffic footage:

- Dataset 1: Our first dataset consists of snapshots of an online traffic camera on the intersection Racecourse Rd. and Boundary Rd. (RB) in Melbourne, published by Vicroads. Since the snapshot used in this study is refreshed every 120 s, our system could only detect certain anomalous behaviors, for example, entering traffic-restricted areas. Hence, this dataset was used only for the detection of vehicles driving on the bicycle lane.
- Dataset 2: Considering the low sampling rate of Dataset 1, we used recorded traffic footage as Dataset 2 that was sampled more frequently from two highways: Panónska cesta (PA1 & PA2, https://www.youtube.com/watch?v=JmFjluIQGJw), M7 Clem Jones Tunnel (M7), and the intersection of Huangshan Rd and Tianzhi Rd in Heifei, China (CN, http://www.openits. cn/openData2/602.jhtml). This dataset contained three videos with lengths of 2, 4, and 4 min, respectively, and a total of 3112 frames. Given the higher resolution, we defined anomalous driving behavior as solid-line crossing, entering traffic-restricted areas, and speeding. For the first two categories, we manually annotated the images to establish ground truth for the evaluation of the detected anomalous driving behaviors.

## 4.2. Results

Several experiments were carried out to evaluate the performance of the proposed system in identifying anomalous driving behaviors. The following sections describe the datasets used in the experiments and the evaluation results for the three anomalous driving behaviors.

### 4.2.1. Solid-Line Crossing and Entering Traffic-Restricted Areas

Table 1 shows the detection accuracy for solid-line crossing and entering restricted zones in Dataset 2. The relatively low recall for "entering-traffic-restricted-areas detection" may be due to the low resolution of online traffic cameras. An instance of anomalous driving behavior is shown in Figure 6, where the anomalous behavior (red rectangle; "CL", line-crossing behavior) is illustrated.

**Table 1.** Detection accuracy of anomalous behaviors.

| Behavior Class | Recall | Precision |
|---|---|---|
| Solid-line crossing detection | 0.889 | 0.865 |
| Entering-restricted-areas detection | 0.730 | 0.964 |

### 4.2.2. Speeding behavior

Figure 7 shows the distribution of the estimated speed for the tracked vehicles, which approximately follows a normal distribution. Therefore, a confidence level can be taken into account dependent on random error estimation through the calculated speed-value uncertainties, which is based on the method of variance propagation that considers the effect of variables' uncertainties on the uncertainty of a function based on them. Note that variance propagation can only evaluate random errors associated with the speed estimation. Moreover, because the ground truth of actual speeds is hard to manually obtain, the speed limit of the road was used as the threshold to classify speeding behavior. In our integrated system, anomalous speed, both too high and too low, was identified by comparing the speed limit of the road with the estimated speed within a high confidence interval, e.g., 68% corresponding to $1\sigma_v$.

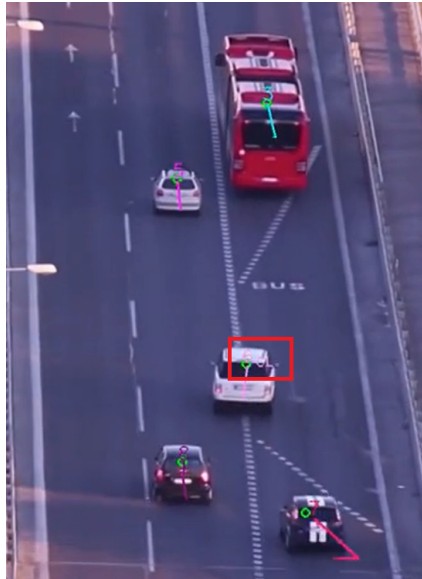

**Figure 6.** Example of detection result (solid-line-crossing behavior).

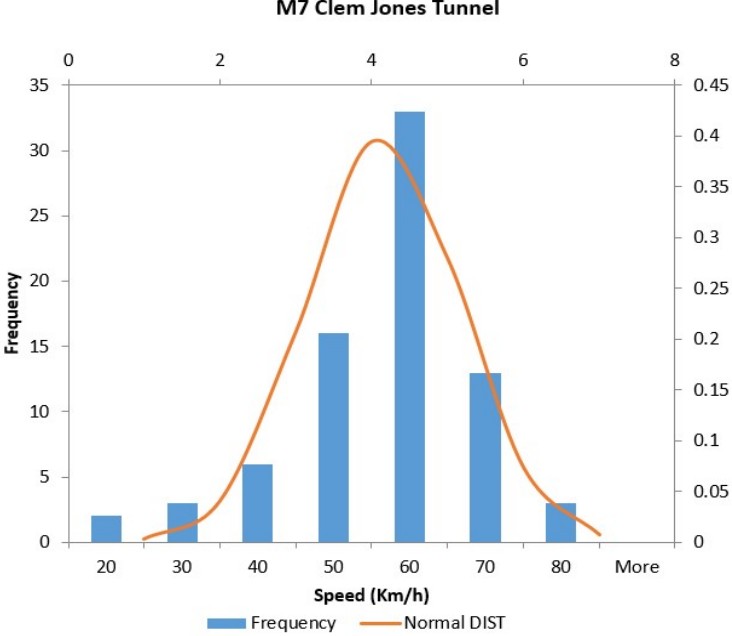

**Figure 7.** Distribution of speed estimations for M7 in Dataset 2.

## 5. Discussion

### 5.1. Combination of Hungarian Algorithm and Kalman Filter

When a vehicle is undetected in successive frames, the Hungarian algorithm assigns the center of other detected vehicles into the trajectory of an undetected vehicle, which causes the wrong trajectory shown in the white line in Figure 8. Incorrect assignment using only the Hungarian algorithm has a large influence on behavior detection and causes many false positives in results, as shown in Table 2. The table concerns two types of footage from the Panónska cesta highway, where the on-site speed limit is 60 km/h. In this situation, a Kalman filter corrects the misassigned center by using a constant-velocity-motion model to predict the supposed center of the undetected vehicle and use it to replace the misassigned center, thus making the trajectory closer to the real situation, as shown by the colored lines in Figure 8.

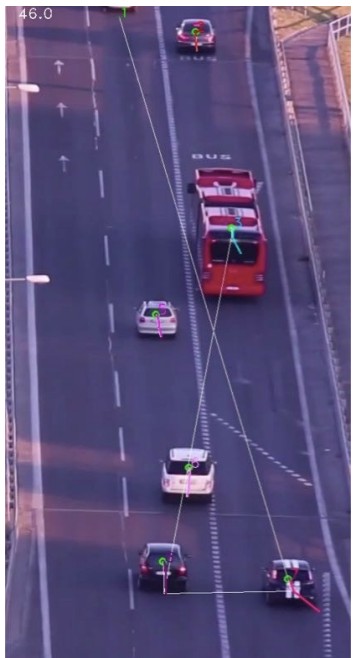

**Figure 8.** Tracking errors caused by the Hungarian algorithm (white lines) can be corrected by a Kalman filter (colored lines).

**Table 2.** Detecting speeding behavior with and without Kalman filter.

| Indicators | Dataset | With Kalman Filter | Without Kalman Filter |
|:---:|:---:|:---:|:---:|
| Mean Speed | Dataset 2-PA1 | 56.66 | 161.98 |
| (km/h) | Dataset 2-PA2 | 62.46 | 57.16 |
| Speed limit within 68% | Dataset 2-PA1 | 68.39 | 85.07 |
| confidence interval (km/h) | Dataset 2-PA2 | 64.24 | 63.23 |
| Number of | Dataset 2-PA1 | 3 | 8 |
| speeding behaviors | Dataset 2-PA2 | 10 | 25 |

*5.2. Integration and Visualization*

The main feature of the system is its ability to integrate online data collection, vehicle detection, vehicle tracking, and behavior classification so that users (urban authorities or road users) can easily obtain the pattern as long as the system has access to a set of traffic footage. The results from the anomalous-driving-behavior detection component are then paired with locations of the footage. Information of anomalous driving behavior at a certain location and time periods can then be analyzed to generate the respective patterns. For example, for our first dataset (VicRoads), the bicycle lane was set as the traffic-restricted area, and when any other objects, such as vehicles and pedestrians, entered this area, it was recorded as anomalous behavior. A bicycle lane with a smaller number of such behaviors can be recommended as a 'cyclist-friendly' route to individuals. Figure 9 shows an example of a hotspot map that can be created based on the observed behaviors.

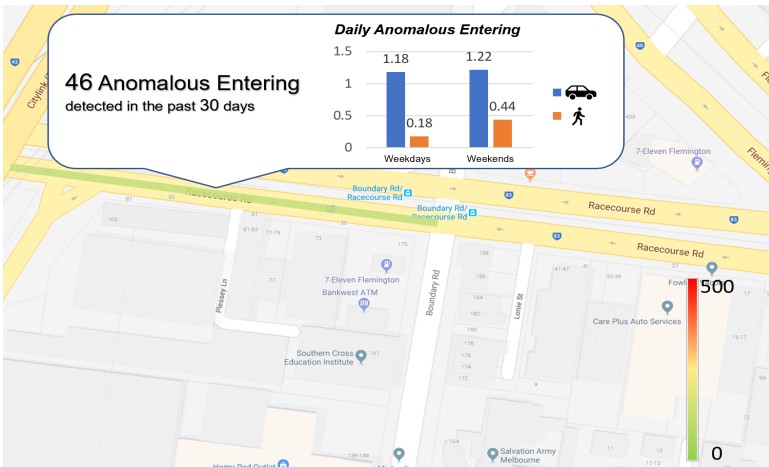

**Figure 9.** Example of hotspot map showing the number of daily anomalous entering on weekdays and weekends.

### 5.3. High Adaptability

This system can be applied to any multiobject-tracking scenarios with only centroids provided, and with a Mask R-CNN model covering wide range of classes of objects. Compared to single-object tracking systems, this system can deal with complex situations with various objects mixing, and remove interference from unnecessary objects. Moreover, the system can be adapted to other functions, for instance, the automatic identification of solid lines and traffic-restricted areas, thus saving the trouble of manually re-setting the geometrical parameters every time for different types of footage.

The detection component in our research runs at approximately 6 s per frame on a PC with an Intel (R) Core (TM) i7-3632QM processor @ 2.20 GHz and 8.0 GB of RAM. For the tracking component, average computation time is approximately 4 s per frame. These computation times can be significantly reduced by running the algorithms on a GPU, thereby enabling the system to perform in near real time and provide data for online applications.

### 5.4. Limitations

Despite our system providing high accuracy and high adaptability, it has several limitations that are summarized as follows:

- In scenarios with high traffic volume where the movement of vehicles is not continuous, the Kalman filter may not be able to correctly track the vehicles due to its simple motion model.
- The location and angle of the camera might cause significant distortion and occlusion in the images and hence affect detection and tracking performance. Theoretically, a bird's-eye view is the most appropriate angle for the camera as it minimizes occlusion. Figure 10 shows an example of a low camera angle where the proposed system cannot perform well due to occlusion.
- In our system, driving behaviors are classified based on simple rules and thresholds, and accuracy may be affected by the threshold setting, which is experimental and subjective. The integration of clustering and classification methods can be used to automatically classify different driving behaviors, thus achieving higher accuracy and efficiency. Such integration has already been proposed to solve the problem of personalized driver-workload inference in Reference [24] and personalized driving behavior prediction in Reference [25].
- System accuracy heavily relies on the accuracy of Mask R-CNN detection. In our current implementation, we used the pretrained model of MS COCO [22]. Fine tuning of this pretrained network using local traffic footage can improve vehicle-detection accuracy. Further improvement can be achieved by incremental learning of the detection model as the system operates and more data become available. Data collected over a long period of time can also be used to detect changes

to road rules. For example, a change of speed limit can be inferred from the statistical distribution of the measured speeds for a large number of vehicles over a long period of time.

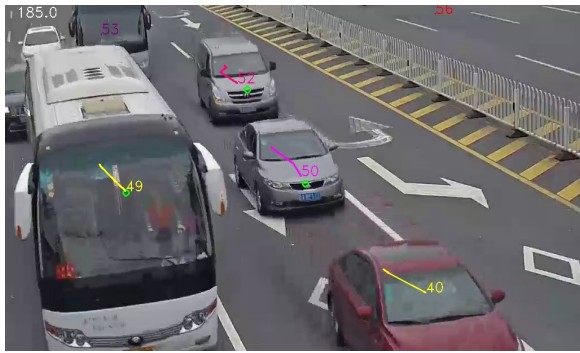

**Figure 10.** Footage with low recording angle.

## 6. Conclusions

In this paper, an integrated system was developed to detect anomalous driving behaviors and visualize the spatial patterns of the distribution of these behaviors using hotspot maps. Overall, the recall and precision rate of the experiment results for solid-lane crossing and entering traffic-restricted areas were around 80%. Regarding speed detection, variance propagation was used to improve system fault tolerance due to the lack of a systematic method to obtain the ground truth of vehicle speeds. Hotspot maps, which were developed as an overview of the number of detected anomalous behaviors during a certain period, can be used as a tool to detect the spatial patterns of such behaviors. In the future, the accuracy of anomalous behavior detections would be improved though methods including the specific training of the Mask-RCNN model, optimization of the default setting matrix of the Kalman filter, and using other cues of each vehicle as preconditions of the Hungarian algorithm (for instance, an assigned vehicle centroid should have same color or same type of vehicle centroids in tracks). Automatic detection of solid lines and traffic-restricted areas can also be achieved, thus saving the trouble of manually re-setting the geometrical parameters every time for different types of footage. Moreover, user interfaces such as hotspot maps would be refined to make it more user-friendly and applicable in a variety of cases.

**Author Contributions:** Conceptualization, E.N., K.K., X.Z., F.W. and W.C.; investigation, X.Z., F.W. and W.C.; software, F.W., W.C. and X.Z.; writing—original-draft preparation, X.Z., F.W. and W.C.; writing—review and editing, E.N. and K.K.; supervision, E.N. and K.K.

**Funding:** This research was funded by the Australian Research Council grant DP170100109.

**Conflicts of Interest:** The authors declare no conflict of interest.

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
