# Peer review of "Show Me a Safer Way: Detecting Anomalous Driving Behavior Using Online Traffic Footage"

_infrastructures, doi:10.3390/infrastructures4020022_

Round 1
Reviewer 1 Report
This paper focus on detecting anomalous driving behavior using traffic cameras available online. Generically, the paper is interesting with a high potential to be improved in further research.
In relation to the presented study, my special concern is related to the speeding anomalous detection. The involved process to compute this indicator, in my opinion, deserves a deeper explanation.
For example, in relation to expression (2) of the variance of the speed of vehicles, it should be explained how is computed (from which observations’ range) the standard deviations of traveling distance and time (σx and σt respectively).
In the table 2 are shown the detecting speeding behavior, how do you find the speed indicators for Dataset1? According to the authors, line 172 to 176, Dataset 1 was only used for the detection of vehicles entering traffic restrict areas , in this case vehicles driving on the bicycle lane.
Author Response
Dear Editor,
We would like to thank the reviewers for their constructive comments. We revised our paper according to the suggested changes that is reflected in the following responses:
Reviewer 1:
Comment: In relation to the presented study, my special concern is related to the speeding anomalous detection. The involved process to compute this indicator, in my opinion, deserves a deeper explanation. For example, in relation to expression (2) of the variance of the speed of vehicles, it should be explained how is computed (from which observations’ range) the standard deviations of traveling distance and time (σx and σt respectively).
Response: the standard deviation of distance is computed by measuring the distance travelled between two consecutive frames across n frames, and for the standard deviation of time a constant value corresponding to 1/2 time interval between consecutive frames is assumed. We have included a detailed explanation of the variance of the speed in Section 3.3.
Comment: In the table 2 are shown the detecting speeding behavior, how do you find the speed indicators for Dataset1? According to the authors, line 172 to 176, Dataset 1 was only used for the detection of vehicles entering traffic restrict areas , in this case vehicles driving on the bicycle lane.
Response: Table 2 contains the results for two datasets of Panónska cesta. To avoid confusion, we have renamed these as Dataset2-PA1/Dataset2-PA2.
Reviewer 2 Report
This work is focused on the automatic detection of anomalous driving behaviors by using the traffic camera that is available online. Various techniques are integrated into the developed framework for this purpose including vehicle detection, vehicle tracking and behavior classification. Different datasets are drawn to validate the developed system. Overall speaking, this is a solid journal paper, which contains some interesting materials worthy publication. The following minor comments are presented to improve the work.
1. It seems the driving behaviors are manually defined via thresholding. Some existing work on driving behavior monitoring or prediction should at least be included such as DOI: 10.1109/TSMC.2017.2764263, DOI: 10.1109/TII.2018.2890141
2. The font size of Figure 1 should be improved.
3. The computation load of the different algorithms should be at least described, e.g. are they suitable for online applications?
4. Can the system be improved by sequentially adapting to/incorporating new labelled data?
Author Response
Dear Editor,
We would like to thank the reviewers for their constructive comments. We revised our paper according to the suggested changes that is reflected in the following responses:
Comment: It seems the driving behaviors are manually defined via thresholding. Some existing work on driving behavior monitoring or prediction should at least be included such as DOI: 10.1109/TSMC.2017.2764263, DOI: 10.1109/TII.2018.2890141
Response: We have included the suggested references in Section 5.4 where we discuss the limitations of the proposed method.
Comment: The font size of Figure 1 should be improved.
Response: The font size in Figure 1 is enlarged and the figure is updated.
Comment: The computation load of the different algorithms should be at least described, e.g. are they suitable for online applications?
Response: Average computation times for the detection and tracking are now discussed in Section 5.3.
Comment: Can the system be improved by sequentially adapting to/incorporating new labelled data?
Response: Yes, we think the performance of the system can be improved by incremental learning from labelled data. This could be an interesting direction for our future research as we have now mentioned in Section 5.4.
Reviewer 3 Report
Very good paper, with a logical structure and deep analysis of the topic, research question is clearly outlined.
Background study is very good, Is it clear what is already known about this topic. Appropriate key studies included, references are recent and relevant. The research question clearly outlined. Data presented in an appropriate way. Used methods are valid and reliable. Results presented in an appropriate way and conclusions supported by results.Topic is highly demanded and definitely will be interesting for the audience.
Author Response
Dear Editor,
We would like to thank the reviewers for their constructive comments. We revised our paper according to the suggested changes that is reflected in the following responses:
Comment: Very good paper, with a logical structure and deep analysis of the topic, research question is clearly outlined. Background study is very good, Is it clear what is already known about this topic. Appropriate key studies included, references are recent and relevant. The research question clearly outlined. Data presented in an appropriate way. Used methods are valid and reliable. Results presented in an appropriate way and conclusions supported by results. Topic is highly demanded and definitely will be interesting for the audience.
Response: Thank you.
Round 2
Reviewer 1 Report
The paper is now in scientific conditions to be published